# Comparison of survival outcomes and anatomically specific severe injuries following traffic accidents among occupants of standard and K-car vehicles: A retrospective cohort study at a teaching hospital in Japan

**Yuko Ono**[1,2]*, **Tasuku Uzawa**[1], **Jun Sugiyama**[1], **Nozomi Tomita**[2], **Takeyasu Kakamu**[3], **Tokiya Ishida**[1], **Joji Kotani**[1], **Kazuaki Shinohara**[2]

1 Department of Disaster and Emergency Medicine, Graduate School of Medicine, Kobe University, Kobe, Japan, 2 Department of Anesthesiology, Ohta General Hospital Foundation, Ohta Nishinouchi Hospital, Koriyama, Japan, 3 Department of Hygiene and Preventive Medicine, School of Medicine, Fukushima Medical University, Fukushima, Japan

* windmill@people.kobe-u.ac.jp

**Data Availability Statement:** All relevant data are within the manuscript and its Supporting

## Abstract

Road traffic accidents are a global health concern. K-car vehicles, also known as "mini vehicles," are defined as those having an engine displacement <660 cc, vehicle length <3.4 m, width <1.48 m, and height <2.0 m. K-cars have become increasing popular not only in Japan but also in other countries. Compared with standard vehicles, the occupant space of a K-car vehicle is considerably smaller; thus, passengers may be more vulnerable to the external forces generated in a collision, which in turn can lead to deformation of the occupant space and less protection against injuries. However, data are scarce regarding whether K-car vehicles are related to poorer survival outcomes and severe trauma. We conducted a retrospective cohort study involving patients who were injured in four-wheeled vehicle accidents between 2002 and 2023 and admitted to a community teaching hospital in Japan. The vehicle configuration was divided into standard and K-car vehicles. The primary endpoint was in-hospital mortality. Other outcomes included severe trauma, defined as Injury Severity Score (ISS) >15, and anatomically specific severe injury of the head and neck, chest, abdomen, pelvis, and extremities, defined as Abbreviated Injury Scale score ≥3. Of 5331 eligible patients, 2384 (44.7%) were K-car vehicle occupants. In propensity score-matched analysis with 1947 pairs, we observed an increase for in-hospital mortality in the K-car vehicle group (2.6% vs. 4.0%, odds ratio 1.53, 95% confidence interval 1.07–2.19). Compared with standard vehicles, K-car vehicles were associated with a greater risk of severe trauma and serious injuries of the head and neck, chest, abdomen, pelvis, and extremities. These study data should be used to encourage vehicle occupants and automobile manufacturers to consider objective facts regarding the safety of vehicles in a traffic accident.

Information files. The minimal anonymized data set used in this study was deposited in the public database and available at: https://doi.org/10.6084/m9.figshare.28138556.v1.

**Funding:** The author(s) received no specific funding for this work.

**Competing interests:** The authors have declared that no competing interests exist.

## Introduction

Road traffic accidents are a global health care concern that places a tremendous economic burden on society. According to the Global status report on road safety 2023, published by the World Health Organization, approximately 1.19 million deaths related to road traffic accidents occurred in 2021 [1]. Road traffic injuries are the leading cause of death among individuals aged 15–29 years and the 12th leading cause of death among all age groups [1]. Recent estimates suggest that the global economic cost of road traffic injuries is USD 1.8 trillion, roughly equivalent to 10%–12% of global gross domestic product [2]. As such, road traffic injuries are a major health care issue.

Vehicle configuration is an important element in road traffic injury fatalities. K-car vehicles, also known as "mini vehicles," comprise the smallest category of Japanese, expressway-legal motor vehicles [3–5]. K-car vehicles have economic benefits such as a low price and low tax rate; therefore, these vehicles have become increasingly popular in Japan, making up more than 40% of all new cars [6]. K-car vehicles have also become popular in other countries because of their environmentally friendly nature, such as having good fuel efficiency and low carbon dioxide gas emissions.

Compared with that in regular vehicles, the occupant space of K-car vehicles is narrower; these vehicles are therefore less likely to absorb the external forces generated in a traffic accident and are more likely to experience deformation of the occupant space, hampering the ability to protect passengers from the energy applied in a collision [7]. Although K-car vehicles are widely distributed, data are scarce regarding whether this vehicle configuration is associated with severe trauma in a motor vehicle accident [4, 5].

Previous studies have reported that K-car vehicles are related to more severe bowel/mesentery and pelvic injuries than regular vehicles. Nevertheless, this finding was limited to drivers who were wearing a seat belt [4], and no other statistical adjustments were made. Another study from Japan indicated that when participants were classified according to seat belt use and airbag deployment, no significant differences in the severity of injuries were found between drivers of K-cars and regular vehicles. [5]. Given these conflicting findings, the link between vehicle configuration and clinical outcomes for individuals injured in a traffic accident remains unclear, and further investigation is needed to clarify this issue [4, 5].

Earlier research on this subject has missed important details like the time of presentation to the emergency department (ED) (i.e., daytime or nighttime), day of presentation (weekday or weekend), occupant seat position (driver's seat, passenger seat, or rear seat), and high-energy trauma (e.g., death in the same compartment, ejection from the automobile) [4, 5]. These variables could be significant confounders [8–12]. Additionally, previous studies have not assessed relevant outcomes such as physiological severities and the need for emergency interventions such as emergency endotracheal intubation (ETI) or emergency surgery, all of which are important trauma care parameters.

In this study, we used our trauma database, in which we prospectively recorded the abovementioned variables, as well as propensity score (PS) matching analysis—a well-established method for mitigating confounding factors in observational studies—to elucidate the relationship between K-car vehicles and clinical outcomes such as in-hospital mortality, anatomically specific severe injuries, physiological severities, and the need for emergency interventions. We hypothesized that K-car vehicles would be independently associated with worse clinical outcomes in trauma patients after a motor vehicle accident.

## Materials and methods

### Study setting and participants

This single center retrospective cohort study was carried out at a community teaching hospital located in a city of northeast Japan. The study facility serves as a regional trauma center and

annually receives more than 1300 patients with varying levels of injury severity. After the study had received approval by the institutional review board of Ohta Nishinouchi Hospital (no. 10_2024) on 19 July 2024, we enrolled all vehicle occupants injured in traffic accidents occurring between 1 January 2002 and 31 December 2023. The committee waived the requirement for patient consent owing to the observational nature of this study, which involved no interventions and was focused on outcomes from routine trauma practice. The opt-out information was provided on the hospital website. We excluded trauma patients who were injured in vehicle accidents involving vehicles other than standard and K-car vehicles, such as pedestrians and occupants of motorcycles, scooters, bicycles, and heavy trucks. Patients with missing data were excluded, and complete data sets were used for all relevant analyses. The retrospective nature of this study predetermined the sample size; therefore, the estimation of statistical power in advance was not possible in this study. Data were collected from a hospital-based electronic trauma database on August 2024. The authors did not access information that could identify individual participants after data collection. The minimal anonymized data set used in this study was deposited in the public database and available at: https://doi.org/10.6084/m9.figshare.28138556.v1. The institutional review board approved sharing of the anonymized data set.

## Exposure

The main exposure in this study was vehicle configuration, dichotomized into standard vehicle and K-car vehicle. A K-car vehicle was defined as one having an engine displacement <660 cc, vehicle length <3.4 m, width <1.48 m, and height <2.0 m, according to the Japanese Road Transport Vehicle Act [6]. The vehicle configuration was defined as a standard vehicle if any of these criteria were exceeded.

## Outcomes

The primary outcome measure of this study was in-hospital mortality in standard vehicle and K-car vehicle groups. The other outcomes of interest were anatomically specific severe injuries defined as those with Abbreviated Injury Scale (AIS) score ≥3 for each anatomical site, and severe injury defined as Injury Severity Score (ISS) score >15. These definitions of severe trauma have been applied in previous studies [4, 5, 7]. The AIS score ranges from 1 (minor injury) to 6 (fatal injury); the ISS is defined as the sum of the squares of the highest AIS in the three most severely injured body regions [13]. We also compared prehospital length of stay (LOS), defined as the time from the emergency call to arrival at the ED; and hospital LOS, defined as the time from hospital admission to hospital discharge or transfer. Additionally, we compared physiological severities and the need for emergency intervention such as ETI or emergency surgery among study participants. Physiological severities were assessed using the Glasgow Coma Scale (GCS) score, systolic blood pressure (SBP), and respiratory rate, all of which were categorized according to the scoring system of the Revised Trauma Score [14]. Coma was defined as GCS score <9 and shock was defined as SBP <90 mmHg. All trauma parameters were scored by a board-certified emergency physician specialized in trauma care (author K.S.) and entered into our trauma database as soon as possible. To minimize bias, the author scoring trauma-related outcomes did not participate in either planning or performing the statistical analyses.

## Covariates

In this study, we captured covariates that are potentially associated with the vehicle configuration and measured outcomes such as patient demographics (age and sex), admission period

(2002–2008, 2009–2015, 2016–2023), season (spring: March–May; summer: June–August; autumn: September–November; winter: December–February), ED presentation time (8:00–16:59, 17:00–23:59, and 24:00–7:59), ED presentation day (weekday or weekend), vehicle factors such as seatbelt (belted, unbelted, and improper seatbelt use), airbag deployment (equipped and deployed, equipped and not deployed, or not equipped), seat position (driver's seat, passenger seat, and rear seat), collision type (frontal collision, lateral collision, rear-end collision, rollover collision, complex collision, other type of collision), and high energy trauma [9–12, 15–18], as defined in the Japan Advanced Trauma Evaluation and Care guideline (e.g., death in the same compartment, ejection from the automobile) [19]. Injuries associated with traffic accidents are representative of unplanned critical conditions that require rapid diagnosis and definitive intervention. In such conditions, presentation at night or on the weekend can adversely affect outcomes owing to lower staffing levels as well as fatigue and disrupted circadian rhythms among staff [11, 12]. More severe traffic accidents tend to occur at night and on weekends [20, 21]. Therefore, we included the time and day of ED presentation as covariates.

## Statistical analysis

Patients' baseline characteristics are described using the median and interquartile range or mean and standard deviation for continuous variables and number and proportion (%) for categorical variables.

First, crude analysis was performed between the standard vehicle and K-car vehicle groups. The Mann–Whitney U test was used to compare continuous variables between the two groups, and the chi-squared test was used to compare categorical variables between groups. Second, to minimize characteristic differences between the standard vehicle group and K-car vehicle group, we carried out a one-to-one PS matching analysis without replacement. By adjusting for differences in measured baseline characteristics, PS matching analysis can mimic the effects of randomization in a randomized controlled trial. Applying PS matching involves first estimating the PS and then matching patients based on their estimated scores. This process is followed by comparing the outcomes among matched patients. Accordingly, each patient in the standard vehicle group was paired with a patient in the K-car vehicle group based on the nearest estimated PS on a logit scale within a specified range (0.2 of the pooled standard deviation of estimated logits). A multivariable logistic regression model was used to predict the probability of assignment to the K-car vehicle group. All categorical variables listed in Table 1 were dummy-coded and included as explanatory variables in the logistic regression. Patients' ages were treated as a continuous variable and were also incorporated into the model. The standardized difference (SD) was used to evaluate the covariate balance; an absolute SD of <0.1 suggests adequate variable balance after PS matching [22]. All statistical analyses were conducted using IBM SPSS Statistics for Windows, version 25.0 (IBM Corp., Armonk, NY, USA). A p-value of <0.05 was considered to indicate statistical significance. Scatter plots were created using GraphPad Prism 9 (GraphPad Software, San Diego, CA, USA).

## Sub-analysis

To assess the robustness of the PS matching analysis mentioned above, the associations between K-car vehicles and outcomes were validated using different statistical assumptions. Accordingly, for each outcome, we conducted inverse probability of treatment weighting (IPTW) analysis and unconditional logistic regression analysis using PS as an explanatory variable. IPTW analysis is used to adjust for confounding in observational studies. For IPTW analysis, we leveraged the PS to balance baseline patient characteristics between the K-car vehicle

**Table 1. Demographic and clinical characteristics of study participants.**

| | Full cohort | | | | PS-matched cohort | | | |
|---|---|---|---|---|---|---|---|---|
| | Standard vehicle (n = 2947) | K-car vehicle (n = 2384) | P | SD (%) | Standard vehicle (n = 1947) | K-car vehicle (n = 1947) | P | SD (%) |
| **Age, y** | | | | | | | | |
| Median (interquartile range) | 39.0 (24.0–57.0) | 46.0 (25.0–63.0) | <0.001 | NA | 42.0 (26.0–58.0) | 42.0 (24.0–61.0) | 0.647 | NA |
| Mean±standard deviation | 40.7 ±20.3 | 44.5±22.1 | <0.001 | 17.9 | 42.7±19.7 | 42.6±22.1 | 0.790 | −0.5 |
| **Sex** | | | <0.001 | | | | 0.654 | |
| Male | 1643 (55.8) | 1119 (46.9) | | −17.9 | 1003 (51.5) | 989 (50.8) | | −1.4 |
| Female | 1304 (44.2) | 1265 (53.1) | | 17.9 | 944 (48.5) | 958 (49.2) | | 1.4 |
| **Admission period** | | | <0.001 | | | | 0.807 | |
| 2002–2008 | 1510 (51.2) | 922 (38.7) | | −25.3 | 845 (43.4) | 833 (42.8) | | −1.2 |
| 2009–2015 | 859 (29.1) | 789 (33.1) | | 8.6 | 606 (31.1) | 625 (32.1) | | 2.2 |
| 2016–2023 | 578 (19.6) | 673 (28.2) | | 20.3 | 496 (25.5) | 489 (25.1) | | −0.9 |
| **Season** | | | 0.643 | | | | 0.886 | |
| Spring (March–May) | 716 (24.3) | 547 (22.9) | | −3.3 | 470 (24.1) | 449 (23.1) | | −2.4 |
| Summer (June–August) | 776 (26.3) | 655 (27.5) | | 2.7 | 535 (27.5) | 540 (27.7) | | 0.4 |
| Autumn (September–November) | 756 (25.7) | 613 (25.7) | | 0 | 492 (25.3) | 502 (25.8) | | 1.1 |
| Winter (December–February) | 699 (23.7) | 569 (23.9) | | 0.5 | 450 (23.1) | 456 (23.4) | | 0.7 |
| **Presentation time** | | | <0.001 | | | | 0.890 | |
| 8:00–16:59 | 1493 (50.7) | 1342 (56.3) | | 11.2 | 1057 (54.3) | 1042 (53.5) | | −1.6 |
| 17:00–23:59 | 903 (30.6) | 686 (28.8) | | −3.9 | 569 (29.2) | 578 (29.7) | | 1.1 |
| 24:00–7:59 | 551 (18.7) | 356 (14.9) | | −10.2 | 321 (16.5) | 327 (16.8) | | 0.8 |
| **Presentation day** | | | 0.002 | | | | 0.528 | |
| Weekday | 2012 (68.3) | 1721 (72.2) | | 8.5 | 1360 (69.9) | 1378 (70.8) | | 2.0 |
| Weekend | 935 (31.7) | 663 (27.8) | | −8.5 | 587 (30.1) | 569 (29.2) | | −2.0 |
| **Seat belt** | | | 0.002 | | | | 0.930 | |
| Belted | 2058 (69.8) | 1768 (74.2) | | 9.8 | 1427 (73.3) | 1436 (73.8) | | 1.1 |
| Unbelted | 813 (27.6) | 561 (23.5) | | −9.4 | 473 (24.3) | 463 (23.8) | | −1.2 |
| Improper seatbelt use | 76 (2.6) | 55 (2.3) | | −1.9 | 47 (2.4) | 48 (2.5) | | 0.6 |
| **Airbag deployment** | | | 0.006 | | | | 0.799 | |
| Equipped and deployed | 1049 (35.6) | 935 (39.2) | | 7.4 | 645 (33.1) | 655 (33.6) | | 1.1 |
| Equipped and not deployed | 947 (32.1) | 681 (28.6) | | −7.6 | 605 (31.1) | 615 (31.6) | | 1.1 |
| Not equipped | 951 (32.3) | 768 (32.2) | | −0.2 | 697 (35.8) | 677 (34.8) | | −2.1 |
| **Seat position** | | | <0.001 | | | | 0.431 | |
| Driver's seat | 1923 (65.3) | 1826 (76.6) | | 25.1 | 1401 (72.0) | 1437 (73.8) | | 4.1 |
| Passenger seat | 521 (17.7) | 366 (15.4) | | −6.2 | 342 (17.6) | 319 (16.4) | | −3.2 |
| Rear seat | 503 (17.1) | 192 (8.1) | | −27.4 | 204 (10.5) | 191 (9.8) | | −2.3 |
| **Collision type** | | | <0.001 | | | | 0.909 | |
| Frontal collision | 1460 (49.5) | 1265 (53.1) | | 7.2 | 1019 (52.3) | 1019 (52.3) | | 0 |
| Lateral collision | 457 (15.5) | 291 (12.2) | | −9.6 | 268 (13.8) | 264 (13.6) | | −0.6 |
| Rear-end collision | 324 (11.0) | 178 (7.5) | | −12.1 | 168 (8.6) | 166 (8.5) | | −0.4 |
| Rollover collision | 305 (10.3) | 366 (15.4) | | 15.3 | 246 (12.6) | 253 (13.0) | | 1.2 |
| Complex collision | 365 (12.4) | 260 (10.9) | | −4.7 | 230 (11.8) | 222 (11.4) | | −1.2 |
| Other type of collision | 36 (1.2) | 24 (1.0) | | −1.9 | 16 (0.8) | 23 (1.2) | | 4.0 |
| **High energy trauma** | | | 0.038 | | | | 0.747 | |
| Yes | 1255 (42.6) | 1083 (45.4) | | 5.6 | 852 (43.8) | 862 (44.3) | | 1.0 |

*(Continued)*

**Table 1.** (Continued)

| | Full cohort | | | | PS-matched cohort | | | |
|---|---|---|---|---|---|---|---|---|
| | Standard vehicle (n = 2947) | K-car vehicle (n = 2384) | P | SD (%) | Standard vehicle (n = 1947) | K-car vehicle (n = 1947) | P | SD (%) |
| No | 1692 (57.4) | 1301 (54.6) | | −5.6 | 1095 (56.2) | 1085 (55.7) | | −1.0 |

Data are expressed as n (%) unless otherwise noted. NA, not available; PS, propensity score; SD, standardized difference.

and standard vehicle groups by weighting each individual in the analysis according to the inverse probability of being assigned to the K-car vehicle group. Logistic regression analysis using PS as an explanatory variable was also repeated in the subgroup of patients who wore a seat belt, those involved in accidents where the airbag was deployed, patients who were involved in a frontal collision, and patients who were sitting in the driver's seat. This method was chosen owing to the small sample size of each subgroup and the potential decrease in statistical power with the PS matching technique.

## Results

### Participant flow

During the study period, 26,620 trauma patients were transported to the study facility. Of those, 11,142 (41.9%) were involved in a traffic accident (**Fig 1**). Among them, 5374 patients were injured in accidents involving vehicles other than a standard and K-car vehicle; 437 patients with missing data were excluded from the analysis. The amount of missing data was small (<4%) for all relevant analyses. The crude analysis included the remaining 5331 patients with complete data sets. Of those, 2384 (44.7%) were K-car vehicle occupants. Using one-to-one PS matching, we selected 1947 pairs of patients with trauma who were involved in an accident with a standard or K-car vehicle. After matching, the distribution of PSs between standard and K-car vehicles was similar (**S1 Fig**).

### Characteristics of study participants

The characteristics of standard vehicle and K-car vehicle groups for both the full cohort and PS-matched cohort are presented in **Table 1**. Compared with the standard vehicle group, patients in the K-car vehicle group were more likely to be older and female patients. There was also a large imbalance in admission periods and presentation times between groups. For example, the K-car vehicle group was more likely to be admitted to the study facility during 2016–2023 and in the daytime (8:00–16:59) and less likely to present during 2002–2008 or in the early morning (24:00–7:59) compared with the standard vehicle group. Patients in the K-car vehicle group were also more likely to wear a seat belt and were more likely to be the driver. The proportion of rollover collisions was higher in the K-car vehicle group but that of rear-end collisions was lower, in comparison with the standard vehicle group. After PS matching, these variable distributions were closely balanced, with all SDs under 10% between the two groups.

### Primary outcomes

For in-hospital mortality, differences between the standard vehicle and K-car vehicle groups are shown in **Fig 2** and **S1 Table**. In PS-matched patients, increased in-hospital mortality was found in the K-car vehicle group (2.6% vs. 4.0%; odds ratio [OR] 1.53; 95% confidence interval [CI] 1.07–2.19). As shown in **Fig 2**, the relationship between increased mortality and K-car

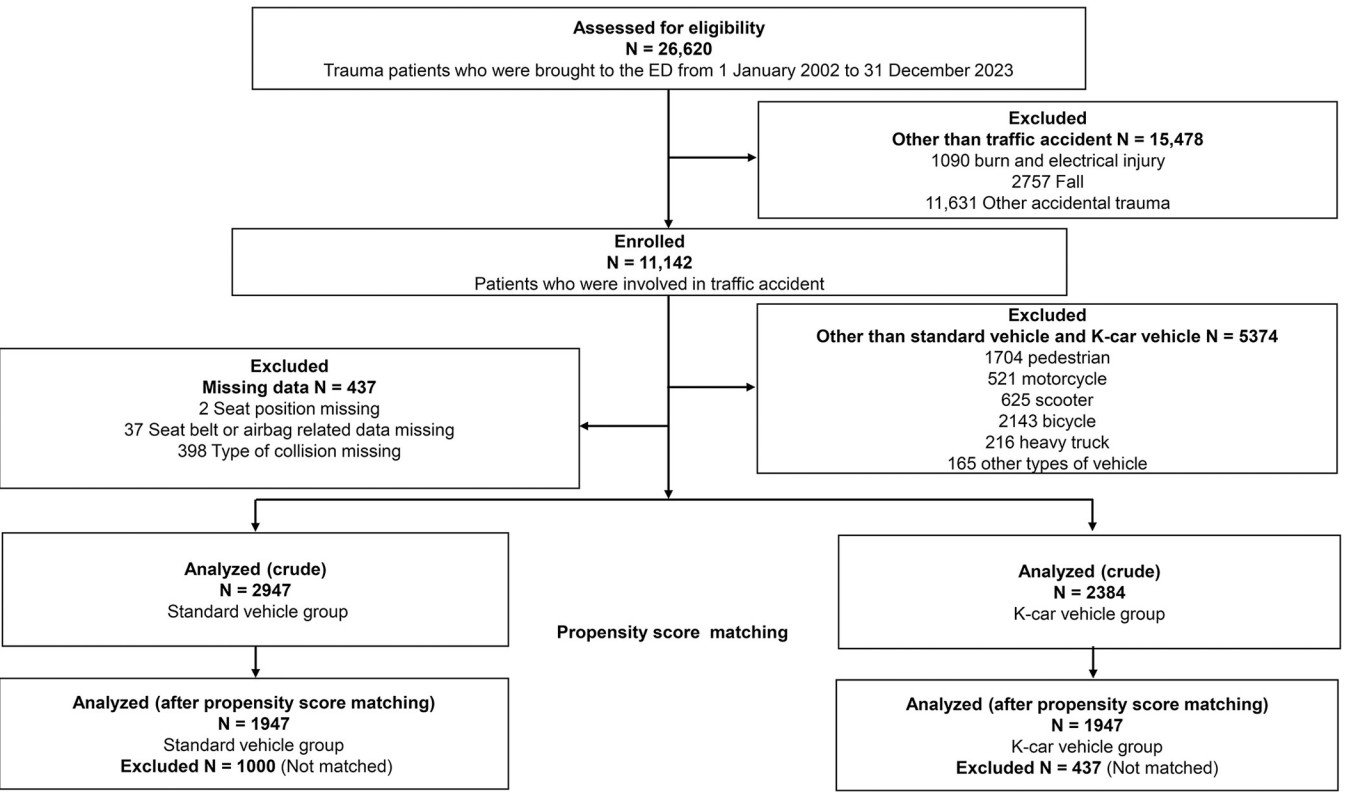

**Fig 1. Flow chart showing the selection process for study participants.** ED, emergency department.

vehicles was also confirmed in other statistical models, including the logistic regression model using PS as an explanatory variable (adjusted OR 1.52; 95% CI 1.10–2.10) and in IPTW analysis (OR 1.57; 95% CI 1.22–2.02). These associations persisted in the any prespecified subgroups of patients (S2 Fig). The increased odds of hospital mortality among K-car vehicle occupants were particularly evident in the subgroup of patients who wore a seat belt, those who were the driver, and those involved in accidents where the airbag was deployed (S2 Fig).

## Other outcomes

Fig 3 and S2 Table show associations between the vehicle type and anatomically specific severe injuries. After PS matching, a greater risk of severe trauma with ISS >15, and a greater risk of severe head or neck, chest, abdominal, and extremity trauma were observed in the K-car vehicle group (Fig 3). The association between increased risks of anatomically specific severe injuries and K-car vehicles was also consistent in all other prespecified statistical models (Fig 3). As shown in S3 Fig, these associations persisted in all prespecified subgroups. The relationship between an increased risk of anatomically severe injuries and K-car vehicles was especially pronounced in the subgroup of patients who wore a seat belt and those involved in accidents where the airbag was deployed (S3 Fig). In PS-matched cohorts, prehospital LOS and Hospital LOS were similar between the standard vehicle and K-car vehicle groups (S4 Fig).

As for physiological severity, increased risk of coma with a GCS score <9 and shock with SBP <90 mmHg were observed in the K-car vehicle group (Fig 4 and S3 Table). The proportion of patients who required emergency ETI and emergency surgery was also significantly higher in K-car vehicle group (Fig 5 and S4 Table).

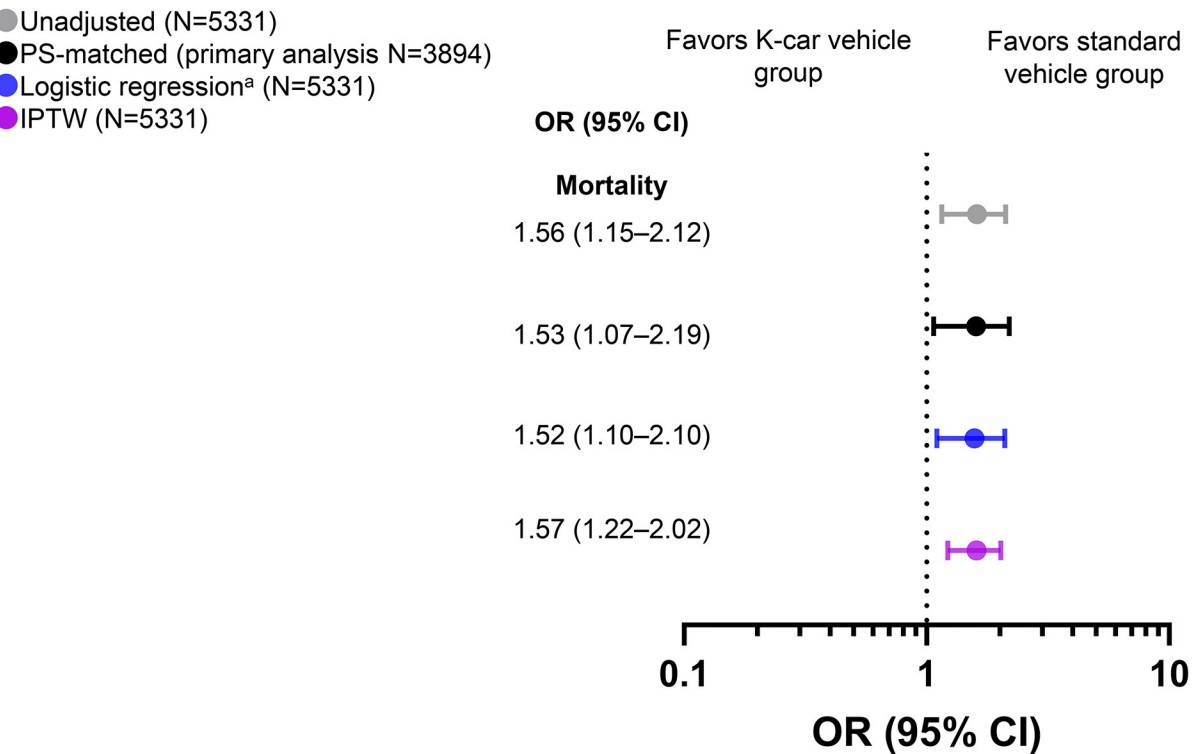

**Fig 2. Odds ratios for in-hospital mortality among study participants: Standard vehicle group versus K-car vehicle group.** The reference set was patients in the standard vehicle group. [a] PS adjustment, as described in the Methods. CI, confidence interval; IPTW, inverse probability of treatment weighting; OR, odds ratio; PS, propensity score.

## Discussion

This retrospective cohort study at a community teaching hospital in Japan found that K-car vehicles were related to increased hospital mortality as well as an increased risk of severe trauma, with ISS >15, in comparison with standard vehicles. Additionally, the occupants of K-car vehicles were more likely to have a serious injury to the head or neck, chest, abdomen, and extremities; this group was also more likely to present to the ED with coma and shock and to need emergency ETI or emergency surgery, compared with occupants of standard vehicles. These associations remained consistent across both the full cohort and PS-matched cohort, as well as under all other prespecified statistical assumptions. Moreover, these associations persisted in all prespecified subgroups of patients such as those who wore a seat belt, those involved in an accident where the airbag was deployed, those who experienced a frontal collision, and patients sitting in the driver's seat. Our results suggest that K-car vehicle occupants require special attention owing to the higher risk of adverse outcomes and potential need for emergency interventions and additional health care resources. Our results will help consumers choose safer vehicle types for purchase and aid the automobile industry in manufacturing safer vehicle bodies.

Our primary analysis showed that the OR of in-hospital mortality was 1.53, meaning that there was an approximately 50% increased risk of death if the traffic accident involved a K-car vehicle versus a standard vehicle. Similarly, our PS matching analysis demonstrated that the ORs for serious injuries of the head and neck, chest, abdomen, pelvis, and extremities were 1.32, 1.20, 1.28, 1.51, and 1.37, respectively. These results suggest an approximately 20%–50% increased risk of a serious injury involving these parts of the body in a K-car vehicle crash

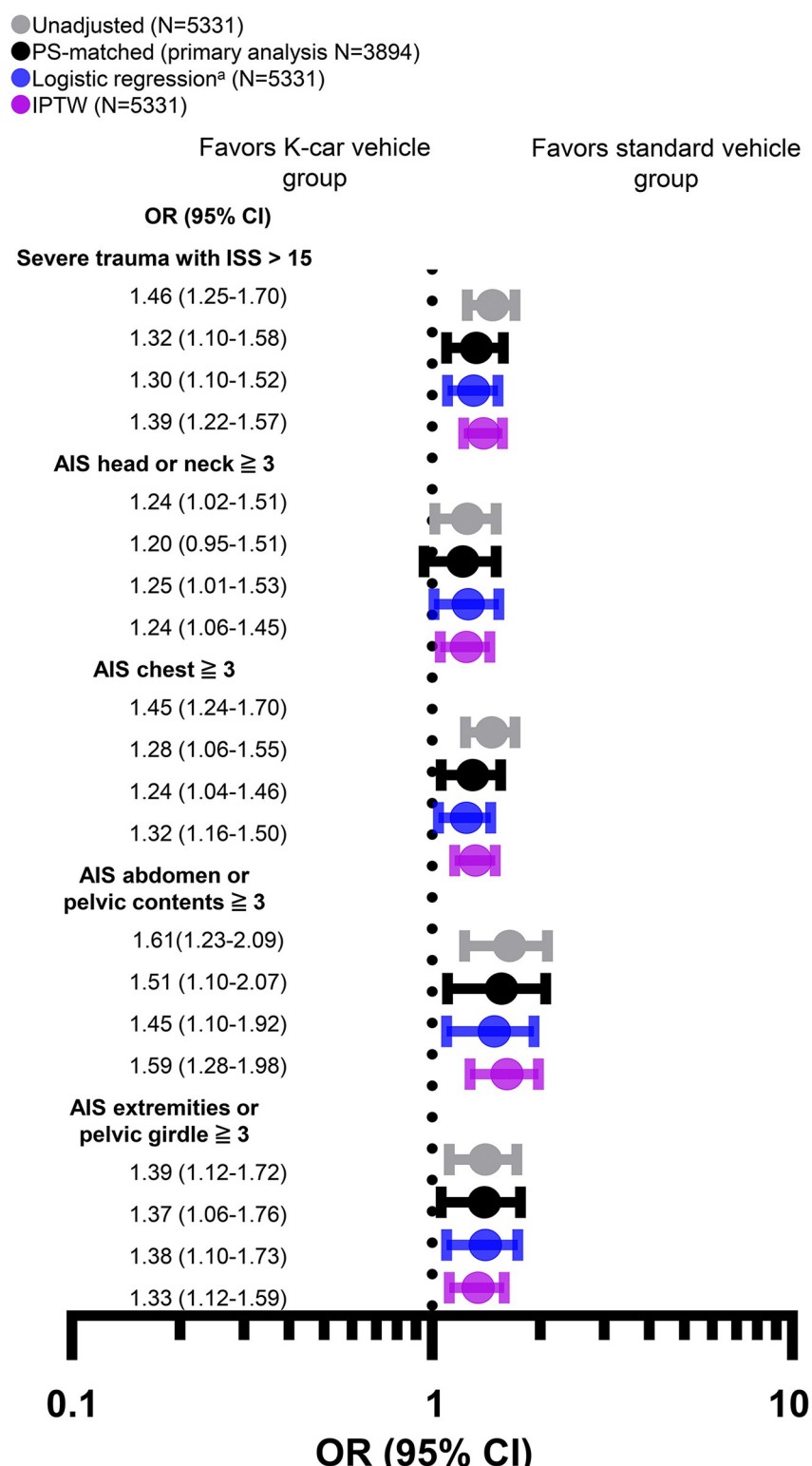

**Fig 3. Odds ratios for severe trauma with ISS >15 and anatomical-site specific severe injury with AIS ≥3 of each body component among participants: Standard vehicle group versus K-car vehicle group.** The reference set was the standard vehicle group. [a]PS adjustment, as described in the Methods. CI, confidence interval; IPTW, inverse probability of treatment weighting; OR, odds ratio; PS, propensity score; AIS, Abbreviated Injury Scale; ISS, Injury Severity Score.

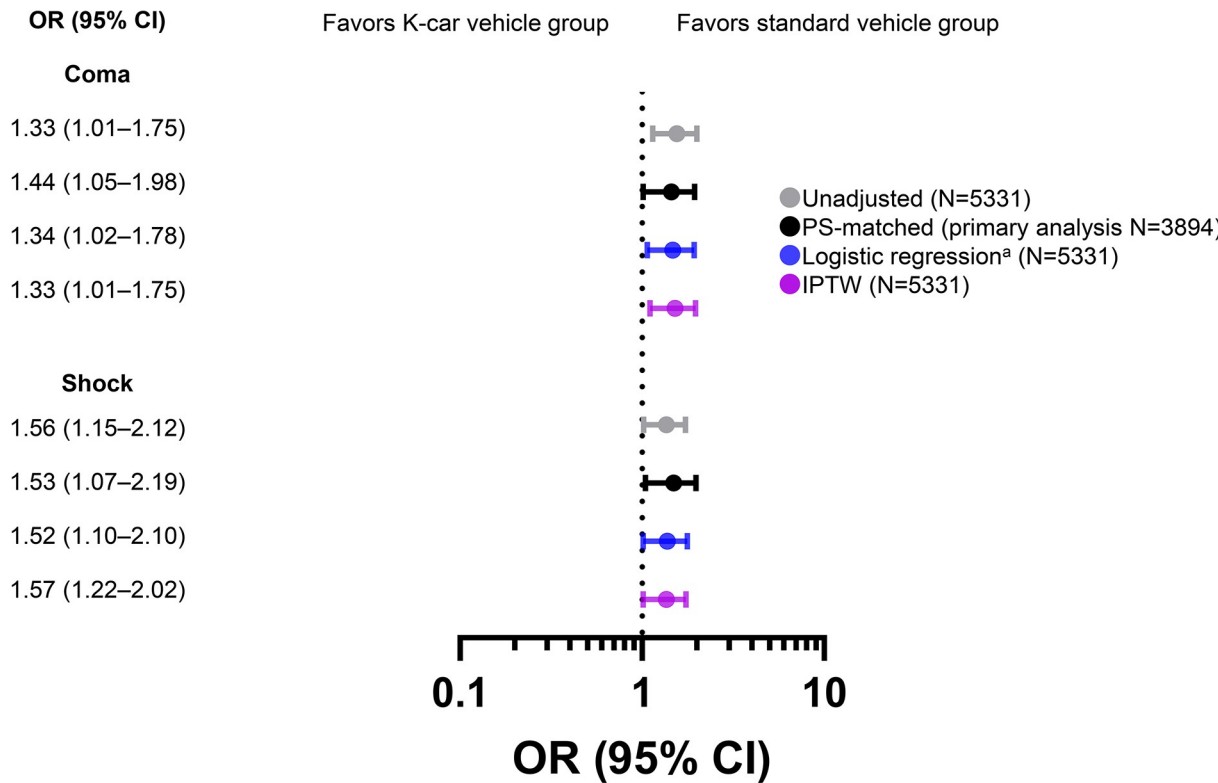

**Fig 4. Odds ratios for coma and shock at emergency department presentation among study participants: Standard vehicle group versus K-car vehicle group.** The reference set was patients in the standard vehicle group. Coma was defined as Glasgow Coma Scale score <9 and shock was defined as systolic blood pressure <90 mmHg. [a]PS adjustment, as described in the Methods. CI, confidence interval; IPTW, inverse probability of treatment weighting; OR, odds ratio; PS, propensity score.

versus accidents in a standard vehicle. We believe that the differences observed in this study are clinically important considering their serious consequences.

Several potential mechanistic reasons could explain our findings regarding more serious injury to the head or neck, chest, abdomen, and extremities among occupants of K-car vehicles versus standard vehicles. First, compared with regular vehicles, the passenger space in K-cars is narrower. This narrower space might reduce the vehicle's ability to absorb energy in a traffic accident, thereby failing to prevent deformation of the occupant space and to adequately protect passengers from the forces of a collision [7]. K-car vehicle occupants may also be more likely to hit the ceiling, windshield, side or other inner components of the vehicle at the time of a collision because of the narrower passenger space, in comparison with that in a standard vehicle. Additionally, K-car vehicles are lighter in weight than regular vehicles. A lighter vehicle may be more likely to become destabilized by the external force generated in a collision, leading to more severe occupant injury [23]. More severe anatomical injury would lead to physiological instability requiring emergency intervention, resulting in poorer survival outcomes. Other potential factors contributing to adverse outcomes among injured K-car vehicle occupants could be linked to variations in socioeconomic status. The purchase price and taxes of some K-car vehicles are much lower than those of a regular vehicle; therefore, the owners of K-car vehicles might have lower socioeconomic status. The association between poorer survival and lower socioeconomic status has been reported [24], although our database does not record these variables. The failure to adjust for socioeconomic status in the current study

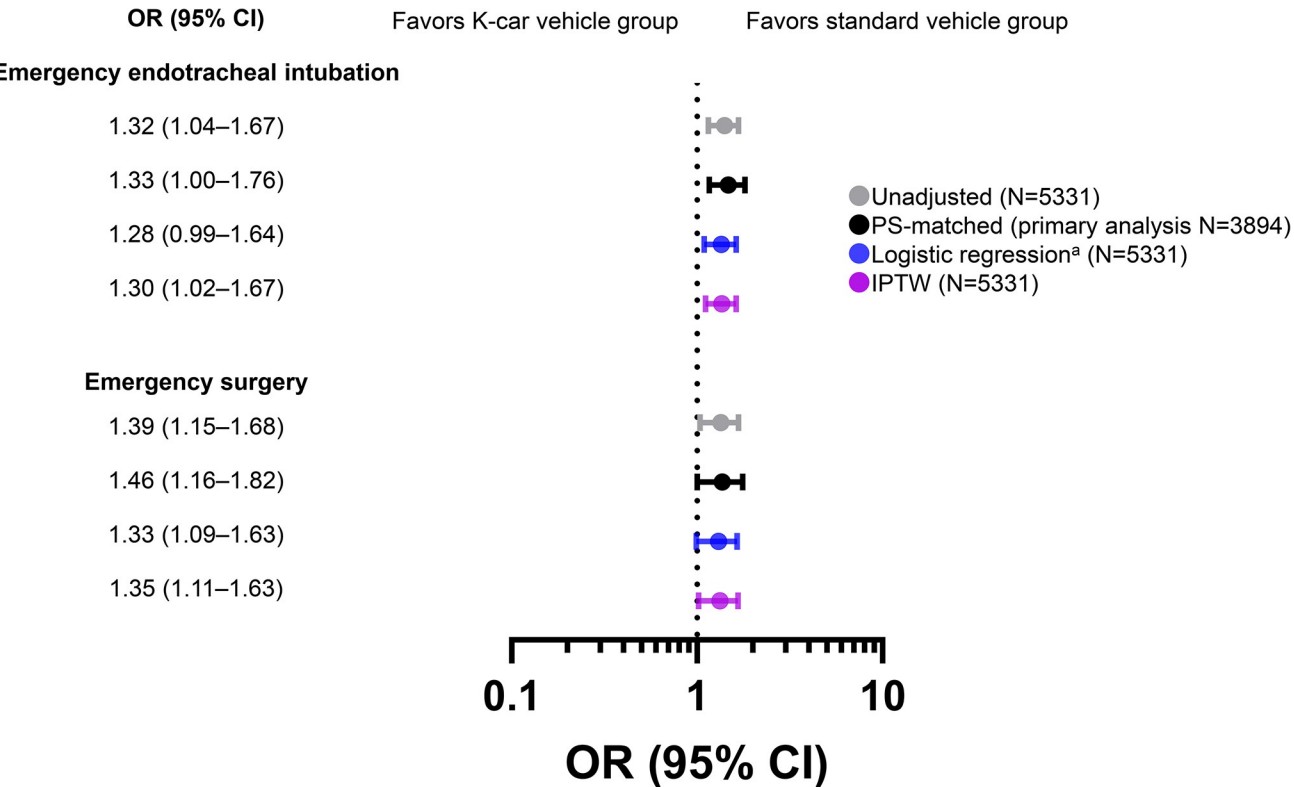

**Fig 5. Odds ratios for emergency interventions among study participants: Standard vehicle group versus K-car vehicle group.** The reference set was patients in the standard vehicle group. [a]PS adjustment as described in the Methods. CI, confidence interval; IPTW, inverse probability of treatment weighting; OR, odds ratio; PS, propensity score.

might therefore have resulted in an apparently stronger association between K-car vehicles and worse outcomes.

Our subgroup analysis showed that the odds of mortality and anatomically severe injuries in K-car vehicles occupants were especially increased in the subgroup of patients who wore a seat belt and those involved in accidents where the airbag was deployed. These results suggest the need for safer restraint systems for K-car vehicle occupants. A previous experimental study found that an optimized safety system, such as one combining a seat belt force limiter, pretensioner, energy absorbing steering system, and knee bolster, was effective in reducing the risk of injury in simulation models of a K-car vehicle driver [25]. Another method to reduce the injury risk to K-car vehicle occupants is increasing the vehicle's rigidity. A previous simulation study found that when a K-car is less rigid, the steering column, instrument panel, and toe pan can hit the chest, knee, and foot of the driver in an experimental collision into a solid wall and large car [25]. According to our real-world data, K-car vehicle occupants were more likely to have severe injuries to the chest and extremities, which is in agreement with those experimental findings. Combining our findings with previous experimental results, we propose that automobile manufacturers invest in producing more rigid K-car vehicles with an optimized restraint system to reduce the risk of severe injury and death.

Unlike our observations, a previous study from Japan reported that after classifying participants according to seat belt use and airbag deployment, there were no significant differences in injury severity among drivers of K-cars and regular vehicles [4]. In another previous study, K-car vehicle occupants tended to have more severe abdominal and pelvic injuries than the

occupants of regular vehicles, but these associations did not reach statistical significance [5]. There are several plausible explanations for these discrepancies. First, our findings are derived from an approximately 10 times larger sample size than those of previous studies [4, 5]. We believe that our study has adequate statistical power to discriminate clinically meaningful differences. After PS matching, our cohort still had a total of 3894 samples. Our post-hoc power calculation using G*Power 3 for Windows (Heinrich Heine University, Dusseldorf, Germany) demonstrated that the estimated power was 0.996 at an α level of 0.05 for the measured primary outcome. Second, in our analysis, we adjusted for several important confounding factors such as time of presentation (daytime or nighttime), day of presentation (weekday or weekend), seat position (driver's seat, passenger seat, or rear seat), and high-energy trauma. Previous studies [4,5] neither recorded nor adjusted for the abovementioned variables. Our study found that the proportions of high energy trauma and rollover collision were higher among K-car vehicles than those for regular vehicles, both of which are associated with greater injury severity [9, 18]. Therefore, failure to adjust for these variables in previous studies could have resulted in even stronger associations between K-car vehicles and severe injury. Third, we divided seat belt status into belted, unbelted, and improper seatbelt use and airbag status into equipped and deployed, equipped and not deployed, and not equipped. In contrast, previous studies have classified seat belt status into only belted and unbelted and airbag status into only deployed or not deployed [4, 5]. Because equipment and airbag deployment as well as proper use of a seatbelt are important factors in the prevention of severe injury and death [15, 17], our classifications and adjustment in the multivariable analysis are of crucial importance. We believe that a combination of these factors might account for the discrepancies between our findings and those of previous research.

Our results concur to some degree with those of previous studies. For example, the distribution of anatomically specific severe injuries with AIS ≥3 and the mortality rate were comparable to those of previous research [4, 5]. For example, in one previous study, the proportions of occupants having an AIS score ≥3 among K-car vehicle and standard vehicle occupants were 10.0% versus 9.5%, 14.5% versus 9.9%, 12.7% versus 11.4%, and 4.5% versus 2.2% for the head or neck, chest, abdomen, and extremities, respectively [4]. In that study, the difference in the mortality rate between these two groups was 4.5% versus 2.2% [4]. However, these results were limited to drivers who were wearing a seat belt, and no other statistical adjustments were made [4]. Our study reinforces these past results by expanding the observations to occupants other than a restrained driver. Our data also corroborate these findings by using a much larger trauma patient sample, with analyses adjusted according to the abovementioned important confounding factors using PS matching analysis.

In Japan, every K-car model must pass crash safety testing prior to being sold. However, our results suggest that K-car vehicles are at a disadvantage in traffic accidents, as compared with regular vehicles, despite passing these safety tests. Our observations provide an opportunity for policy makers to reconsider updating the safety regulations for K-car vehicles to promote better outcomes among traffic accident victims.

We believe our findings have several implications for the consumer, medical personnel, and automobile manufacturers. First, our study offers consumer safety information to help with vehicle selection during purchase. Previous research on vehicle preferences among teenagers in the United States indicates that price is an important factor, leading to a preference for mini-vehicles, small vehicles, and older models [26, 27]. Informing consumers about the relatively higher risks associated with K-car vehicles compared with regular vehicles could influence their choices. Second, our findings could be useful in prehospital settings for patient transfer. If the vehicle involved in an accident is a K-car, prehospital medical personnel can anticipate that the occupant may have severe internal injuries (e.g., pelvic, abdominal, or chest

trauma) even if external injuries appear minor. In our study, these findings were consistent regardless of seatbelt use and airbag deployment. This information could help prehospital medical personnel to prioritize transfer of K-car vehicle occupants to a tertiary medical center. Our findings could also be helpful for hospital emergency medical teams. If the injured occupants are involved in a K-car vehicle accident, the emergency medical team can prepare for emergency interventions such as ETI, surgery, and blood transfusion before hospital arrival. Third, our findings should motivate automobile manufacturers to invest in enhanced safety technologies for K-car vehicles.

## Limitations and strengths

The current study has several limitations. First, because this was a single-site observational study, the generalizability of the findings might be limited. For example, our study setting was a tertiary emergency medical facility in Japan. Generally, patients with severe injuries, but not those with minor injuries, are more likely to be transferred to such a facility. Thus, the patient sample in this study may be skewed in terms of injury severity. Nevertheless, we believe that the associations between vehicle configuration and severe injuries are unlikely to be significantly biased because the selection of patients transported to our hospital was based on injury severity, regardless of vehicle type. For the same reason, we believe that a Japanese community hospital setting would not strongly bias the relationship between vehicle configuration and poorer survival outcomes.

Second, even though we made rigorous adjustments in the PS-matched analysis, other unmeasured factors may have confounded our results, as with any observational study. For instance, several important covariates, such as vehicle velocity, social status, insurance status, use of vasopressors, and alcohol use at the time of injury [7, 8, 24, 28, 29] were not recorded in our database. Additionally, environmental factors such as rain, snow, road surface (wet, snow-covered, icy), and fog can affect the injury severity [30, 31]. Our database did not capture these variables. Nevertheless, whereas environmental factors are potential predictors of severe injuries, they were not related to the vehicle configuration in this study. For instance, environmental factors could not be theoretically used to determine whether the study participant was in a standard or K-car vehicle. Therefore, these factors do not necessarily meet the criteria for confounding according to the classical epidemiologic framework. We therefore believe that environmental factors would not severely bias the relationship between vehicle configuration and poorer survival outcomes. Further study is required to clarify whether adjustment for these variables can affect the associations between vehicle configuration and injury severity examined in this study.

Despite these limitations, we believe that the current study has several strengths. First, our database captures relevant trauma parameters such as physiological severity and the need for emergency intervention, such as emergency ETI or emergency surgery. Previous studies have not assessed these outcomes. Second, this is the first study to use PS matching analysis for this research question, with adjustment for several important confounding factors that previous studies have not measured. Additionally, our sample size was large, allowing for subgroup analysis that focused on important trauma subsets. We therefore believe that this study provides more robust and specific results than those of previous studies. Third, the amount of missing data was small (<4%) in all the analyses, minimizing the selection bias and maximizing the quality of the PS matching analysis. Fourth, the measured outcomes were objective (i.e., hospital mortality and hospital LOS), making them less susceptible to diagnostic errors. All parameters in this study were entered into the database prospectively. Furthermore, to reduce the risk of biased assessment, the author who constructed the database (K.S.) was not

involved in any of the statistical analysis. Therefore, we believe that the current study accurately depicted the impact of vehicle configuration on mortality and anatomically specific injury severity among patients involved in traffic accidents.

## Conclusions

In this retrospective observational study, we found that K-car vehicle accidents were associated with increased mortality, as compared with standard vehicle accidents. We also observed that occupants of K-car vehicles were more likely to have severe trauma with an ISS score >15 and were more likely to have severe injuries to the head or neck, chest, abdomen, and extremities than standard vehicle occupants. Moreover, occupants of K-cars were more frequently admitted to the ED with coma and shock and had a greater need for emergency ETI and emergency surgery. We hope that these findings will motivate vehicle occupants and automobile manufacturers to consider objective safety facts regarding injury in a traffic accident.

## Supporting information

**S1 Fig.** Distribution of propensity scores in the unmatched (A) and matched groups (B).
(PPTX)

**S2 Fig. Subgroup analysis of hospital mortality among study participants: Standard vehicle group versus K-car vehicle group.** The reference set was patients in the standard vehicle group. [a]PS adjustment, as described in the Methods. CI, confidence interval; OR, odds ratio; PS, propensity score.
(PPTX)

**S3 Fig. Subgroup analysis of anatomical severities among study participants: Standard vehicle group versus K-car vehicle group.** The reference set was the standard vehicle group. [a]PS adjustment, as described in the Methods. CI, confidence interval; ISS, Injury Severity Score; OR, odds ratio; PS, propensity score.
(PPTX)

**S4 Fig. Prehospital and hospital LOS among injured patients: Standard vehicle group versus K-car vehicle group.** (A, B) Prehospital LOS in the full (A) and PS-matched (B) cohort. (C, D) Hospital LOS in the full (C) and PS-matched (D) cohort. Prehospital LOS is defined as time from the emergency call to ED arrival. Hospital LOS is defined as time from hospital admission to hospital discharge or transfer. representing the data distribution (circles), mean (horizontal bar), and interquartile range (vertical bar), respectively. The p-values are derived from the Mann–Whitney U-test. LOS, length of stay; PS, propensity score; ED, emergency department.
(PPTX)

**S1 Table. Comparison of mortality rate: Standard vehicle versus K-car vehicle.**
(DOCX)

**S2 Table. Comparison of anatomical injury severities: Standard vehicle versus K-car vehicle.**
(DOCX)

**S3 Table. Physiological severity among study participants.**
(DOCX)

**S4 Table. Requirement for emergency intervention among study participants.**
(DOCX)

## Acknowledgments

We thank our colleagues at Ohta Nishinouchi Hospital for their earnest trauma practice. We also thank Nozomi Ono, MD (Department of Psychiatry, Hoshigaoka Hospital, Koriyama, Japan) for providing assistance in reviewing the manuscript. Finally, we thank Analisa Avila, MPH, ELS, of Edanz (https://jp.edanz.com/ac) for editing a draft of this manuscript.

## Author Contributions

**Data curation:** Yuko Ono, Tasuku Uzawa, Nozomi Tomita.

**Formal analysis:** Yuko Ono, Tasuku Uzawa, Nozomi Tomita, Takeyasu Kakamu.

**Investigation:** Yuko Ono, Tasuku Uzawa, Jun Sugiyama, Takeyasu Kakamu.

**Methodology:** Yuko Ono.

**Project administration:** Tokiya Ishida, Joji Kotani, Kazuaki Shinohara.

**Resources:** Tokiya Ishida, Kazuaki Shinohara.

**Supervision:** Tokiya Ishida, Kazuaki Shinohara.

**Validation:** Takeyasu Kakamu, Joji Kotani.

**Visualization:** Yuko Ono.

**Writing – original draft:** Yuko Ono.

**Writing – review & editing:** Tasuku Uzawa, Jun Sugiyama, Nozomi Tomita, Tokiya Ishida, Joji Kotani, Kazuaki Shinohara.

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
