## [Decision Letter · Decision Letter 0]

11 Dec 2024

PONE-D-24-51846Comparison of survival outcomes and anatomically specific severe injuries following traffic accidents among occupants of standard and K-car vehicles: a retrospective cohort study at a teaching hospital in JapanPLOS ONE

Dear Dr. Ono,

Thank you for submitting your manuscript to PLOS ONE. After careful consideration, we feel that it has merit but does not fully meet PLOS ONE’s publication criteria as it currently stands. Therefore, we invite you to submit a revised version of the manuscript that addresses the points raised during the review process.

We look forward to receiving your revised manuscript.

Kind regards,

Dr. Satabdi Mitra, M.D(Community Medicine )

Academic Editor

PLOS ONE

Journal Requirements:

Reviewers' comments:

Reviewer's Responses to Questions

**Comments to the Author**

1. Is the manuscript technically sound, and do the data support the conclusions?

Reviewer #1: Yes

Reviewer #2: Yes

Reviewer #3: Yes

2. Has the statistical analysis been performed appropriately and rigorously? 

Reviewer #1: Yes

Reviewer #2: Yes

Reviewer #3: Yes

3. Have the authors made all data underlying the findings in their manuscript fully available?

Reviewer #1: Yes

Reviewer #2: Yes

Reviewer #3: Yes

4. Is the manuscript presented in an intelligible fashion and written in standard English?

Reviewer #1: Yes

Reviewer #2: Yes

Reviewer #3: Yes

5. Review Comments to the Author

Reviewer #1: This document is technically sound and the data supports the conclusions. Statistical analysis appear appropriate and executed correctly. I recommend the following minor corrections:

line 69. typo. Missing "to" in "More likely to experience...

line 82. why is time and day of presentation important here? maybe add a short line on that.

Reviewer #2: This article focuses on comparison of commonly used cars in Japan, and their impact on the survival rates in accidents.

The results clearly indicates the K-car has more mortality rate compared to standard cars.

this results can be further utilized for the safety measures that has to be taken for future.

Reviewer #3: Manuscript Number: PONE-D-24-51846

Manuscript Title: Comparison of survival outcomes and anatomically specific severe injuries following traffic accidents among occupants of standard and K-car vehicles: a retrospective cohort study at a teaching hospital in Japan

The research seems to offer novel insights derived from a retrospective cohort study that examines survival outcomes and particular injury patterns in occupants of standard vehicles compared to K-car vehicles in Japan. The study's emphasis and approach are unique, utilizing propensity score matching along with various statistical models, thereby enhancing the analytical rigor. Below my comment:

1. Originality of Research

• The study addresses a critical gap in understanding the safety outcomes of K-car vehicles compared to standard vehicles. By utilizing a large sample size and robust statistical methods, it contributes novel insights to the field of traffic injury prevention.

• One recommendation is to elaborate further on the context of existing literature. Specifically, the discussion could benefit from a clearer comparison of the findings with previous studies, emphasizing the added value of this research.

2. Results Reporting

• The manuscript explicitly mentions that the findings have not been published elsewhere, which maintains the novelty of the study. However, it would be helpful for the authors to explicitly clarify in the introduction or discussion that this is the first study to combine propensity score matching with subgroup analyses for this research question.

3. Technical Standards and Detail

• The propensity score matching is rigorously implemented and enhances the reliability of comparisons between vehicle types. Including scatter plots of propensity score distribution (S1 Figure) adds transparency.

• While the manuscript addresses the statistical methods comprehensively, additional detail on how missing data (4%) were handled could further strengthen confidence in the findings. For instance, was imputation considered, or were only complete cases analyzed?

4. Conclusions and Data Support

• The conclusions are directly supported by the data, and the manuscript thoughtfully discusses the implications for consumers, emergency responders, and automobile manufacturers.

• One suggestion is to explore more actionable recommendations for automobile manufacturers, such as specific design improvements to mitigate the risks identified in K-cars.

5. Presentation and Language

• The manuscript is generally well-written and easy to follow. However, certain technical terms, such as "propensity score matching" and "inverse probability of treatment weighting," could benefit from brief lay explanations for readers less familiar with these methodologies.

• Simplifying the descriptions in the introduction for a broader audience might help increase accessibility to a wider readership.

6. Ethics and Research Integrity

• The ethical approvals and processes for obtaining informed consent are clearly described. The study adheres to ethical standards for retrospective research.

• A minor recommendation is to specify whether the IRB considered risks related to data sharing, even though all data were anonymized.

7. Reporting Guidelines and Data Availability

• The manuscript adheres to community standards for reporting and data availability. The inclusion of supporting tables and figures aligns well with transparency and reproducibility standards.

• To improve accessibility, the authors could consider depositing the anonymized dataset in a publicly available repository with links provided in the manuscript.

General comments

• Figures and Tables: While figures effectively illustrate key findings, the legends could provide more interpretative insights, such as clarifying the clinical significance of the odds ratios presented.

• Discussion Depth: The authors could explore broader policy implications, such as the need for updated safety regulations for K-cars, especially considering their rising popularity.

• Subgroup Analysis: Subgroup analyses (e.g., by gender or age group) are insightful but could be further contextualized in terms of their potential policy or clinical implications.

• Address unmeasured confounders: Consider discussing the impact of unmeasured variables (e.g., environmental conditions) more extensively in the limitations.

• Provide raw data access: Ensure supporting data are accessible for reproducibility.

• Language refinement: Review for minor grammatical improvements to enhance clarity.

The evaluation indicates that this manuscript is suitable for acceptance, pending minor revisions.

6. PLOS authors have the option to publish the peer review history of their article (what does this mean?). If published, this will include your full peer review and any attached files.

Reviewer #1: No

Reviewer #2: No

Reviewer #3: No

---

## [Author Response · Author response to Decision Letter 0]

17 Jan 2025

Point-by-point responses to the reviewers’ comments

To Reviewer #1: 

We greatly appreciate your time and effort in reviewing our manuscript and providing your insightful comments. We have incorporated the recommended changes into the revised (R1) version of our manuscript. Any changes based on your comments are indicated in red.

Reviewer #1: This document is technically sound and the data supports the conclusions. Statistical analysis appears appropriate and executed correctly. I recommend the following minor corrections:

Response:

We greatly appreciate the reviewer's positive evaluation of the manuscript. Thank you very much.

line 69. typo. Missing "to" in "More likely to experience...

Response:

We corrected this typological error in accordance with your comment. Thank you for your careful review of the manuscript.

line 82. why is time and day of presentation important here? maybe add a short line on that.

Response:

Thank you for noting this point. Injuries associated with traffic accidents are representative of unplanned critical conditions that require rapid diagnosis and aggressive intervention. Under such conditions, night and weekend presentation can adversely affect outcomes owing to lower staffing levels as well as staff fatigue and disrupted circadian rhythms (cited references 11, 12). Furthermore, most severe traffic accidents tend to occur at night and on weekends (cited references 20, 21). We therefore included emergency department presentation time and day as covariates. A corresponding description is given in in the text (R1 version, page 10, line 155 to page 11, line 161). 

To Reviewer #2: 

Thank you very much for taking the time to review our manuscript. Our point-by-point responses are provided below:

Reviewer #2: This article focuses on comparison of commonly used cars in Japan, and their impact on the survival rates in accidents. The results clearly indicate the K-car has more mortality rate compared to standard cars. These results can be further utilized for the safety measures that has to be taken for future.

Response:

We thank the reviewer very much for this positive evaluation of our manuscript.

To Reviewer #3: 

Thank you very much for taking the time to review our manuscript and your pertinent suggestions. In accordance with your advice, we have amended our manuscript. Any changes based on your comments are indicated in red.

Reviewer #3: Manuscript Number: PONE-D-24-51846

Manuscript Title: Comparison of survival outcomes and anatomically specific severe injuries following traffic accidents among occupants of standard and K-car vehicles: a retrospective cohort study at a teaching hospital in Japan

The research seems to offer novel insights derived from a retrospective cohort study that examines survival outcomes and particular injury patterns in occupants of standard vehicles compared to K-car vehicles in Japan. The study's emphasis and approach are unique, utilizing propensity score matching along with various statistical models, thereby enhancing the analytical rigor. Below my comment:

Response:

We really appreciate the reviewer's positive evaluation of the manuscript. Thank you very much.

1. Originality of Research

• The study addresses a critical gap in understanding the safety outcomes of K-car vehicles compared to standard vehicles. By utilizing a large sample size and robust statistical methods, it contributes novel insights to the field of traffic injury prevention.

Response:

We are pleased to see this constructive comment. Thank you very much.

• One recommendation is to elaborate further on the context of existing literature. Specifically, the discussion could benefit from a clearer comparison of the findings with previous studies, emphasizing the added value of this research.

Response:

We appreciate the reviewer’s insightful comment. We have amended the discussion based on this advice (R1 version, page 29, line 349 to page 31, line 387).

The added value of this research

We believe that the value added by this study to the existing medical literature mainly comprises the following two points. First, our findings are derived from an approximately 10 times larger sample size than those of previous studies. This large sample size allowed us to discriminate clinically meaningful differences in the mortality rate and anatomically specific severe injuries between the standard vehicle and K-car vehicle groups. The large sample size also facilitated subgroup analysis focusing on important trauma subsets, such as patients who wore a seat belt, those who were the driver, those involved in accidents where the airbag was deployed, and patients involved in a frontal collision, among others. Thus, we believe that our study provides more specific results and suggestions than previous studies. 

Second, in our analysis, we adjusted for several important confounding factors such as the time of presentation (daytime or nighttime), day of presentation (weekday or weekend), seat position (driver’s seat, passenger seat, or rear seat), and high-energy trauma, using propensity score (PS) matching methods. Previous studies (cited references 4,5) neither recorded nor adjusted for the above-mentioned variables. Thus, we believe that our findings provide more robust results than those of previous studies. 

Other strengths of this manuscript are listed in the section on strengths and limitations (R1 manuscript, page 34, line 436 to page 35, line 451). We believe that these descriptions in the R1 manuscript will help readers to understand the added value of this research.

Comparison of the findings with previous studies

There are some discrepancies between current and previous studies (cited references 4,5). For example, past studies have failed to find statistically significant differences in mortality and anatomically specific injury severities between standard and K-car vehicle groups (cited references 4,5). We have provided a potential explanation for how these discrepancies arose (R1 manuscript, page 29, line 349 to page 31, line 375). 

However, our results concur to some degree with those of previous studies (cited references 4,5). For example, the distribution of anatomically specific severe injuries with Abbreviated Injury Scale scores ≥3 and the mortality rate in this study were comparable to those of previous research. We have explained how this study reinforces the results of some previous studies (page 31, lines 376 to 387). We believe that these descriptions will help readers to compare our study with previous research.

Thank you for this pertinent advice, which has prompted us to provide a more thorough discussion. 

2. Results Reporting

• The manuscript explicitly mentions that the findings have not been published elsewhere, which maintains the novelty of the study. However, it would be helpful for the authors to explicitly clarify in the introduction or discussion that this is the first study to combine propensity score matching with subgroup analyses for this research question.

Response:

We agree with the reviewer’s suggestion. Accordingly, we have incorporated the following text into the strengths and limitations section of the discussion (R1 version, page 34, lines 439 to 443):

This is the first study to use PS matching analysis for this research question, with adjustment for several important confounding factors that previous studies have not measured. Additionally, our sample size was large, allowing for subgroup analysis that focused on important trauma subsets. We therefore believe that this study provides more robust and specific results than those of previous studies.

3. Technical Standards and Detail

• The propensity score matching is rigorously implemented and enhances the reliability of comparisons between vehicle types. Including scatter plots of propensity score distribution (S1 Figure) adds transparency.

Response:

We thank the reviewer very much for this positive evaluation of the manuscript.

• While the manuscript addresses the statistical methods comprehensively, additional detail on how missing data (4%) were handled could further strengthen confidence in the findings. For instance, was imputation considered, or were only complete cases analyzed?

Response:

Thank you for raising this point. Patients with missing data were excluded, and complete data sets were used for all relevant analyses. Corresponding descriptions are now given in the R1 version of the manuscript (page 8, lines 107 to 108). 

4. Conclusions and Data Support

• The conclusions are directly supported by the data, and the manuscript thoughtfully discusses the implications for consumers, emergency responders, and automobile manufacturers.

Response:

We are grateful for this constructive comment. Thank you very much.

• One suggestion is to explore more actionable recommendations for automobile manufacturers, such as specific design improvements to mitigate the risks identified in K-cars.

Response:

Thank you for this pertinent suggestion. Our subgroup analysis showed that the odds of mortality and anatomically severe injuries in K-car vehicle occupants were especially increased in patients who wore a seat belt, those who were the driver, and those involved in accidents where the airbag was deployed (S2 Fig and S3 Fig). These results suggest the need for safer restraint systems for K-car vehicle occupants. A previous experimental study found that optimized safety systems such as those combining a seat belt force limiter, pretensioner, energy-absorbing steering system, and knee bolster, were effective in reducing the injury risk in simulation models of K-car vehicle drivers (cited reference 25).

Another method to reduce the injury risk to K-car vehicle occupants is increasing the vehicle rigidity. A previous study reported that when a K-car is less rigid, the steering column, instrument panel, and toe pan can hit the chest, knee, and foot of the driver in an experimental collision into a solid wall and large car (cited reference 25). In our real-world data, K-car vehicle occupants were more likely to have severe chest and extremities injury (Fig.3), which is in agreement with the above experimental findings. 

Combining our findings with previous results, we propose that automobile manufacturers produce more rigid K-car vehicles with an optimized restraint system to reduce the risk of severe injury. We have added a corresponding description in the R1 version of the manuscript (page 28, line 334 to page 29, line 348). 

5. Presentation and Language

• The manuscript is generally well-written and easy to follow. However, certain technical terms, such as "propensity score matching" and "inverse probability of treatment weighting," could benefit from brief lay explanations for readers less familiar with these methodologies.

Response:

Thank you very much for this positive evaluation of the manuscript. In accordance with this advice, we have added brief explanations of the PS matching and inverse probability of treatment weighting (IPTW) methods in the R1 version of the manuscript, as follows:

Page 11, line 171 to page 12, line 175

By adjusting for differences in measured baseline characteristics, PS matching analysis can mimic the effects of randomization in a randomized controlled trial. Applying PS matching involves first estimating the PS and then matching patients based on their estimated scores. This process is followed by comparing the outcomes among matched patients. 

Pag 13, lines 193 to 196:

IPTW analysis is used to adjust for confounding in observational studies. For IPTW analysis, we leveraged the PS to balance baseline patient characteristics between the K-car vehicle and standard vehicle groups by weighting each individual in the analysis according to the inverse probability of being assigned to the K-car vehicle group.

• Simplifying the descriptions in the introduction for a broader audience might help increase accessibility to a wider readership.

Response:

In accordance with this advice, we removed redundant portions in the introduction (page 5, lines 60 to 62 of the original version of the manuscript).

6. Ethics and Research Integrity

• The ethical approvals and processes for obtaining informed consent are clearly described. The study adheres to ethical standards for retrospective research.

Response:

We are pleased to see this positive evaluation of the manuscript.

• A minor recommendation is to specify whether the IRB considered risks related to data sharing, even though all data were anonymized.

Response:

In accordance with the reviewer’s opinion, we queried the institutional review board of Ohta Nishinouchi Hospital regarding this point. The board scrutinized the data file and approved sharing of this anonymized data set. This information was added to the R1 version of the manuscript (page 8, lines 114 to 115).

7. Reporting Guidelines and Data Availability

• The manuscript adheres to community standards for reporting and data availability. The inclusion of supporting tables and figures aligns well with transparency and reproducibility standards.

Response:

Thank you very much for this positive evaluation of the manuscript.

• To improve accessibility, the authors could consider depositing the anonymized dataset in a publicly available repository with links provided in the manuscript.

Response:

In accordance with your advice, we have deposited the minimal anonymized data set to FigShare. The dataset is now available at: https://doi.org/10.6084/m9.figshare.28138556.v1.

This information was added to the R1 version of the manuscript (page 8, lines 112 to 115).

General comments

• Figures and Tables: While figures effectively illustrate key findings, the legends could provide more interpretative insights, such as clarifying the clinical significance of the odds ratios presented.

Response:

We agree with the reviewer’s comment and have revised the manuscript accordingly. Our primary analysis showed that the odds ratio (OR) for in-hospital mortality was 1.53 (Fig. 2), meaning that there was an approximately 50% increased risk of death if the traffic accident involved a K-car vehicle, as compared with a standard vehicle crash. Likewise, our PS matching analysis demonstrated that the ORs of serious injuries to the head and neck, chest, abdomen, pelvis, and extremities were 1.32, 1.20, 1.28, 1.51, and 1.37, respectively (Fig. 3). This means that there was an approximately 20%–50% increased risk of serious injury for each body part if involved in a K-car vehicle crash, in comparison with standard vehicle accidents. We believe that these observed differences are clinically important, considering their serious consequences. The corresponding text was incorporated into the R1 version of the manuscript (Page 27, lines 307 to 314).

• Discussion Depth: The authors could explore broader policy implications, such as the need for updated safety regulations for K-cars, especially considering their rising popularity.

Response:

We agree with the reviewer’s comment. As you noted, our observation provides an opportunity for policy makers to reconsider updating the safety regulations for K-car vehicles to promote better outcomes among traffic accident victims. We have amended the manuscript in accordance with this comment (R1 version, page 31, line 388 to page 32, line 392).

• Subgroup Analysis: Subgroup analyses (e.g., by gender or age group) are insightful but could be further contextualized in terms of their potential policy or clinical implications.

Response:

Thank you for this valuable suggestion. In our subgroup analysis, the odds of mortality and anatomically severe injuries among K-car vehicles occupants were especially increased in patients who wore a seat belt, those who were the driver, and those involved in accidents where the airbag was deployed (S2 Fig and S3 Fig). These results suggest that safer restraint systems for K-car vehicle occupants are required (R1 manuscript, page 28, line 334 to page 29, line 340). Regarding this issue, please also see our point-by-point responses to your comment #4.

• Address unmeasured confounder

---

## [Editor Report · Decision Letter 1]

21 Jan 2025

Comparison of survival outcomes and anatomically specific severe injuries following traffic accidents among occupants of standard and K-car vehicles: a retrospective cohort study at a teaching hospital in Japan

PONE-D-24-51846R1

Dear Dr. Yuko Ono,

We’re pleased to inform you that your manuscript has been judged scientifically suitable for publication and will be formally accepted for publication once it meets all outstanding technical requirements.

Kind regards,

Satabdi Mitra, M.D(Community Medicine )

Academic Editor

PLOS ONE
---

## [Editor Report · Acceptance letter]

28 Jan 2025

PONE-D-24-51846R1 

PLOS ONE

Dear Dr. Ono, 

I'm pleased to inform you that your manuscript has been deemed suitable for publication in PLOS ONE. Congratulations! Your manuscript is now being handed over to our production team.

Kind regards, 

on behalf of

Dr Satabdi Mitra 

Academic Editor

PLOS ONE